# Anemia, micronutrient deficiency, and elevated biomarkers of inflammation among women and children in two districts in the Northern Region of Ghana: A pilot study

Seth Adu-Afarwuah[1]*, Sika M. Kumordzie[2,3], K. Ryan Wessells[2,3],
Charles D. Arnold[2], Xiuping Tan[2], Ahmed Fuseini[1], Stephen A. Vosti[2,4],
Marjorie J. Haskell[2,3], Emily R. Becher[2], Jennie N. Davis[2], Katherine P. Adams[2,3],
Reina Engle-Stone[2,3]

1 Department of Nutrition and Food Science, University of Ghana, Legon, Accra, Ghana, 2 Institute for Global Nutrition, University of California, Davis, California, United States of America, 3 Department of Nutrition, University of California, Davis, California, United States of America, 4 Department of Agricultural and Resource Economics, University of California, Davis, California, United States of America

* sadu-afarwua@ug.edu.gh

## Abstract

### Introduction

Previous data had shown a high prevalence of anemia and various micronutrient deficiencies (MNDs) in the Tolon and Kumbungu districts. We aimed to reassess these outcomes among lactating women (LW), women of reproductive age (WRA), and preschool children (PSC) to inform the design of a MN-fortified bouillon cubes trial, including the choice of micronutrients and selection of study sites; among WRA and PSC, we aimed to identify factors associated with anemia, MND, and inflammation, and to examine anemia co-occurring with MND and inflammation.

### Methods

In this cross-sectional study (Nov 2020–Jan 2021), we randomly selected 7 urban and 7 rural clusters per district from those accessible at the time, recruited participants through a random walk-based search, and collected and analyzed breast-milk (LW) and venous blood (WRA and PSC). Ferritin (WRA and PSC), and retinol, retinol-binding protein, and zinc (PSC only) were adjusted for inflammation. Binary outcomes were defined using accepted cut-offs. Data analyses involved descriptive statistics, generalized linear mixed models (identify factors), and Rao-Scott chi-squared test (examine co-occurrence).

**Data availability statement:** Consistent with policy of the Bill & Melinda Gates Foundation, all de-identified data have been made available online at https://osf.io/t3zrn/files/osfstorage.

**Funding:** This work was supported, in whole or in part, by a grant from Helen Keller International (66504-UCD-01), through support from the Bill & Melinda Gates Foundation (INV-007916), to the University of California, Davis. Under the grant conditions of the Foundation, a Creative Commons Attribution 4.0 Generic License has already been assigned to the Author Accepted Manuscript version that might arise from this submission. The funders had no role in study design, data collection and analysis, decision to publish, or preparation of the manuscript.

**Competing interests:** The authors have declared that no competing interests exist.

**Abbreviations:** AGP, α1-acid glycoprotein; CRP, C-reactive protein; Hb, hemoglobin; Hh, Household; HFIAS, Household Food Insecurity Access Scale; ID, iron deficiency; IDA, iron deficiency anemia; LW, lactating women; HAZ, height-for-age z-score; MND, micronutrient deficiency; PSC, preschool aged children; RBP, retinol binding protein; SF, serum ferritin; SR, serum retinol; sTfR, soluble transferrin receptor; SSB, sugar-sweetened beverage; WHZ, weight-for-height z-score; WRA, women of reproductive age.

## Results

Approximately 240 each of LW (average age, 30 y), WRA (30 y), and PSC (41 months) participated. Among LW, 41% had low breastmilk B-12; 8% had low breastmilk retinol. Among WRA, anemia was 31%; prevalences of MNDs were: iron, 45%; zinc, 79%; vitamin A, 1%; B-12, 12%; and folate, 12%; with 15% elevated α1-acid glycoprotein (AGP) or C-reactive protein (CRP). Among PSC, anemia was 36%; prevalences of MNDs were: iron, 57%; zinc, 67%; vitamin A, 19%; and B-12, 19%; with 39% elevated AGP or CRP. Improved source of drinking water was associated with lower odds of anemia, iron deficiency, and vitamin A deficiency in WRA; rural or Kumbungu residence was linked to higher odds of vitamin B-12 or vitamin A deficiency in PSC. Anemia co-occurred with iron deficiency and inflammation in WRA and PSC.

## Conclusion

Anemia and MNDs were common in this setting, suggesting the need for targeted interventions.

## Introduction

Anemia and micronutrient (vitamin and mineral [1]) deficiencies (MNDs) are prevalent, particularly in women and under-five children in many low-income settings [2, 3]. Globally, an estimated 30% of non-pregnant, non-lactating women of reproductive age (WRA) 15–49 y and 40% of preschool children (PSC) aged 6–59 months had anemia in 2019 [4]. More alarming, an estimated 69% of WRA, equivalent to 1.2 billion, were deficient in folate, iron, or zinc, while 56% of PSC, equivalent to 372 million, were deficient in vitamin A, iron, or zinc in 2022 [5]. Anemia and MNDs have debilitating consequences, including increased morbidity and mortality rates, and reduced work capacity in WRA, and poor growth [6–8], cognitive impairment [6, 9], and lower school performance [10] in PSC, with long-term negative effects on educational achievement and economic potential [2].

In Ghana, an estimated 41% of WRA and 49% of PSC were anemic in 2022 [11], while the prevalence rates of vitamin A, vitamin B-12, folate, iodine, iron, and zinc deficiencies remain a matter of public interest for both target groups. In the 2017 Ghana Micronutrient Survey [12, 13], the estimated national prevalence of specific MNDs was 2% for vitamin A, 59% for folate, and 20% for iron among WRA, and 13% for vitamin A and 30% for iron among PSC. Subsequent analysis [14] suggested that nearly 33% of anemia in WRA and at least 19% of anemia in PSC may be attributable to iron deficiency.

While there are reliable statistics in Ghana for anemia [11] and MNDs [13] among WRA and PSC at the national and regional levels, the extent of these deficiencies in many sub-regional or local areas remains largely unknown. The Ghana Micronutrient Survey [12, 13] revealed that the Tolon and Kumbungu districts in the Northern Region were among "pockets" in the country with higher prevalences of anemia

and iron deficiency in WRA and anemia, iron deficiency, and vitamin A deficiency in PSC than elsewhere in the same region. Consequently, our research group suggested that these two districts might be a suitable site for a planned randomized trial of micronutrient-fortified bouillon cubes for preventing anemia and MNDs in Ghana and similar settings in West Africa. Identifying factors associated with anemia, MNDs, and inflammation biomarkers, as well as evaluating their co-occurrence, might help inform strategies to improve nutrition and health outcomes in this setting.

In a pilot study in the Tolon and Kumbungu districts, we aimed to reassess the prevalence of anemia and MNDs among lactating women (LW), WRA, and PSC to inform the planned micronutrient-fortified bouillon intervention trial. The present analysis focused on the micronutrients including vitamin A, vitamin B-12, folate, iodine, iron, and zinc, which were likely to be of public health importance [15] in Ghana and were intended for inclusion in the micronutrient-fortified bouillon cubes [16]. Specifically, we aimed to assess breastmilk micronutrient concentration among LW; in WRA and PSC, we aimed to (a) determine the prevalence of anemia, MND, and elevated biomarkers of inflammation, (b) identify community-, household-, and individual-level factors associated with anemia, MNDs, and inflammation, and (c) evaluate whether anemia co-occurred with MNDs or inflammation significantly more frequently than expected by chance.

## Methods

### Study design, setting, and participants

This was a cross-sectional study. Most people in the Tolon and Kumbungu districts were semi-subsistence farmers [17], and the poverty level in the area was high [18]. The local diet consisted mainly of staples, including maize, millet, sorghum, cowpea, and groundnuts. Each of the district capitals, Tolon and Kumbungu, had reasonably good public amenities, such as treated water, electricity, mobile telephone service, a health facility, and a tarred road from Tamale, the regional capital. However, access to improved source of drinking water and sanitation was not universal across the districts [18].

Potential participants were (i) non-pregnant LW 15−49 y of age with infants 4−18 months of age, (ii) non-pregnant, non-lactating WRA (15−49 y), and (iii) PSC (24−59 months). Exclusion criteria were severe illness warranting hospital referral, body temperature >38 ºC or reported COVID-19 exposure, chronic severe medical condition (e.g., malignancy) or congenital anomaly requiring frequent medical attention, inability to provide informed consent due to impaired decision-making abilities, and current participation in a clinical trial.

The study protocol was approved by the Ghana Health Service Ethics Review Committee (GHS-ERC: 012/07/20) and the University of California, Davis, Institutional Review Board (IRB Number 1536100−1).

### Study procedures

**Participant recruitment.** We obtained the list of "urban" and "rural or semi-rural" clusters [19] in the study area from the Ghana Statistical Service and randomly selected 7 of each per district (total 28 clusters). In the Tolon district, clusters were selected from those that were accessible, as a river had flooded and made some areas difficult to enter. Within each cluster, we aimed to recruit 6–9 participants for each target group (LW, WRA, and PSC) or 18–27 participants in total based on the target sample size. From 09 November 2020 to 16 January 2021, field workers identified eligible households and potential participants through a random walk method, which involved (i) randomly choosing a "starting point" within the cluster, (ii) approaching the "next nearest house" to assess household eligibility and recruit eligible participant(s), and (iii) continuously recruiting eligible participants from the "next nearest house" until the target number of participants had been recruited [20,21]. A household was defined as a group of people who recognized the same head of household, lived together, and shared living expenses and meals. Field workers chose a starting point within a cluster by following a walking route in a randomly selected direction from the center to the outskirt of the cluster where the houses could be easily counted, and selecting one of those houses using a random number generator [22].

At each household approached, field workers assigned an ID number and obtained oral consent from the household head to determine, by using a brief questionnaire, whether the household included any potentially eligible participants. This questionnaire included questions about COVID-19 exposure; if the respondent had a body temperature >38 ºC or reported COVID-19 exposure by a household member, the interview was discontinued and household members advised to seek medical attention. Eligible household members were recruited after field workers explained the study procedures to them (LW and WRA) or their caregivers (PSC), answered any questions, and obtained informed written consent. Depending on the number of participants already recruited in a cluster, we prioritized recruiting individuals from households with eligible participants from two or more target groups. We recruited a participant or participants from one household per house or compound, and one participant per target group per household.

## Data and biological sample collection

At recruitment, field workers used pre-tested questionnaires and electronic tablets to collect information on demographic and socio-economic status (household size, number of children <5 y, household members' educational level, food security [Household Food Insecurity Access Scale (HFIAS) score [23]], main source of drinking water [24], and type of toilet facility [25]). To guide the future planned clinical trial of multiple micronutrient-fortified bouillon cube, we determined household consumption of bouillon over the prior 30 days through a household questionnaire based on the Fortification Assessment Coverage Tool (FACT) developed by the Global Alliance for Improved Nutrition [26], and estimated participants' bouillon "apparent intake" through the Adult Male Equivalent method [27].

Additional individual-level information included participants' age, education, consumption patterns (typical week's servings) of selected food groups, use of micronutrient-containing supplements and micronutrient powders in the past 30 d, and morbidity in the past 7 d. The food groups (by a food frequency questionnaire, FFQ) included fruits, vegetables, sweetened snacks (e.g., biscuits, candies or chocolates), salty snacks (e.g., crisps and salty crackers), and sugar-sweetened beverages (SSBs, e.g., soft drinks and sugar-sweetened fruit drinks). The FFQ was adapted from the World Health Organization STEPS (STEPwise approach to non-communicable disease risk factor surveillance) instrument [28]. Showcards with photos of local food items and typical servings sizes were used as a prompt during interviews. A serving of fruits or vegetables was defined as 80 g [28]. For SSBs, a serving was defined as 120 mL, and for salty snacks and sweetened snacks, each serving was defined as 20 g [29].

On the day after recruitment (or the following working day, if recruitment occurred on a Friday) we invited participants to a central mobile laboratory for anthropometric measurements and biological sample collection. WRA were asked to fast for at least 8 hours before the laboratory visit. After arrival, study staff asked about participants' and their household members' exposure to COVID-19, and measured participants' body temperature. Participants with fever or reported diarrhea, either currently or within the past 24 hours, were deferred with the possibility of returning after their symptoms resolved. For participants without these symptoms, trained anthropometrists used procedures described by the World Health Organization [30] to measure their weight to the nearest 0.05 kg (Seca 874; Seca) and height to the nearest 0.1 cm (Seca 217; Seca). PSC's mid-upper arm circumference was measured to the nearest 0.1 cm (Shorr Productions).

The biological sample collection involved breastmilk from LW, and spot urine and blood from WRA and PSC. From the LW, the study nurse collected ~10 ml of breastmilk using the "full milk sample" method [31] (i.e., taking sample from the breast that was 'more full') and recording the time of day and time since the last breastfeeding. Before collecting the samples, women were asked to breastfeed their children or express breastmilk for 60 seconds from the breast that was sampled. Immediately following collection, breastmilk fat content was measured using the Creamatocrit (CreamatocritPlus Centrifuge; EKF Diagnostics) to centrifuge the milk in a microhematocrit tube to separate the aqueous and fat layers. Subsequently, the length of each layer was measured to calculate the fat content as % of volume, which was then converted to grams per liter (g/L) using a validated regression equation [32,33]. A breastmilk aliquot was stored at <−20° C for retinol and vitamin B-12 analysis.

From WRA and PSC, spot urine samples were collected, placed in a cool box, and later stored at −20°C. Trained phlebotomists collected venous blood (~ 6 mL) into evacuated, trace element-free, serum tubes (Sarstedt AG & Co, Numbrecht, Germany) using standard procedures, and immediately measured hemoglobin (Hb) by Hemocue (Hemoue 301, Hemocue AB Angelholm, Sweden) and malaria antigenemia by a rapid diagnostic test, RDT (BioZEK, B.V., Apeldoorn, Netherlands). For erythrocyte folate measurement in WRA, hematocrit was measured using the GCH-24 Hematocrit centrifuge (Globe Scientific, Mahwah, New Jersey, USA) and in addition, 100 μL of whole blood was aliquoted into polypropylene cryovials containing 1 mL of 1% ascorbic acid solution. The blood samples in the serum blood collection tubes were placed on cold packs and allowed to clot for at least 30 minutes, but not more than 8 hours, before centrifugation at 1,097 x $g$ (3100 RPM) for 10 min. Serum samples were aliquoted and stored at −86 °C [16] for analysis.

Phlebotomists were unable to obtain venous blood samples from 8/224 (4%) WRA and 70/241 (29%) PSC after two attempts and, thus, obtained finger-prick samples for Hb (g/L) and malaria assessments only. Information necessary to interpret biomarker values, including the time of the participant's most recent meal, was recorded at blood draw. Participants received written copies and interpretation of Hb and malaria RDT results; those with low Hb (<120 g/L for WRA; < 110 g/L for PSC) or positive malaria RDT were referred to the nearest health center.

## Laboratory analyses

Breastmilk, whole blood, and serum samples were air-freighted on dry ice for analysis. Breastmilk vitamin A (retinol, nmol/g fat) [34] was measured at the University of California, Davis by using high-performance liquid chromatography (HPLC; Roche Diagnostics, Indianapolis, USA). Breastmilk and serum vitamin B-12 (pmol/L) were measured at the USDA Western Human Nutrition Research Center by using IMMULITE 1000 solid-phase automated competitive binding chemiluminescent enzyme immunoassay (Siemens Healthcare Diagnostics, Duluth, GA, USA) [35]. Serum ferritin (SF, μg/L), soluble transferrin receptor (sTfR, mg/L), retinol-binding protein (RBP, μmol/L), alpha-1-acid glycoprotein (AGP, g/L), and C-reactive protein (CRP, mg/L) were measured at the VitMin Laboratory (Willsteadt, Germany) using a sandwich ELISA technique [36]. Additional analyses included serum retinol (SR, μmol/L) [37], by HPLC at the University of California, Davis; serum zinc (μg/dL) [38], by inductively coupled plasma optical emission spectrometry (ICP-OES) at the University of California, San Francisco MLK Cores Research Facility; whole blood and serum folate (nmol/L) [39], by the microbiological method at the Centers for Disease Control and Prevention; and urinary iodine concentration (UIC, μg/L) by modification of the Sandell-Kolthoff reaction [40] at the University of Ghana. All the participating laboratory analyses had quality control procedures for monitoring the validity of the tests, including appropriate standard reference materials and quality control standards, which were analyzed with all sample runs.

## Outcome variables

Among LW, the continuous outcome measures were breast milk retinol and vitamin B-12 concentrations. We defined low breast milk retinol as <28 nmol/g fat [41]. Currently, there is no universally accepted cut-off for low vitamin B-12 in breast milk [42]. A cut-off of <362 pmol/L exists but is not widely used due to potential overestimations of inadequacy, as it was developed using older laboratory methods that may not have measured breast milk vitamin B-12 accurately, and was based on a relatively small number of samples [43, 44]. Therefore, we defined low breast milk B-12 concentration using two cut-offs, namely <362 pmol/L [43, 44] and <221 pmol/L often used for serum/plasma vitamin B-12 [45].

For WRA and PSC, the continuous outcome variables were blood Hb concentration, indices of iron status (SF, sTfR, and body iron stores (BIS, estimated from SF and sTfR concentrations [46])), iodine status (UIC), vitamin status (serum B-12, erythrocyte folate, serum folate, serum retinol, and RBP), zinc status (serum zinc), and biomarkers of inflammation (AGP and CRP). We used the correction approach described by the Biomarkers Reflecting the Inflammation and Nutritional Determinants of Anemia (BRINDA) project [47] to adjust SF, retinol (PSC only), RBP (PSC), and zinc (PSC) concentrations for inflammation (AGP and/or CRP) before any analysis.

The binary outcomes were anemia or low Hb (including mild, moderate, and severe anemia) [48] and those defining iron deficiency (low SF [49], elevated sTfR [36], and low BIS [46]), iron deficiency anemia (low Hb with low SF) [50], zinc deficiency (low serum zinc) [51], vitamin A deficiency (low serum retinol [52] and low RBP [53]), vitamin B-12 deficiency (low serum B-12 [45]), folate deficiency ("insufficient" [54, 55] or low RBC folate [55, 56] and low serum folate [55, 56]), elevated AGP [57], and elevated CRP [57]. The cut-offs used for the binary outcomes are shown in S1 Table.

**Potential factors associated with anemia, MNDs, and elevated biomarkers of inflammation among WRA and PSC**

Potential independent variables for anemia, ID (by ferritin), zinc deficiency, vitamin A deficiency (by serum retinol), B-12 deficiency, folate deficiency (by RBC folate in WRA only) and inflammation (elevated AGP or CRP) among WRA (7 outcomes) and PSC (6 outcomes) were derived from participants' background characteristics, including *household-level* variables (such as demographic characteristics; household head's level of education; asset index; food security and access scale score; source of drinking water; type of toilet facility; and bouillon cubes consumption) and *individual-level* variables (such as age; marital status (WRA only); educational level or level in school; employment status; typical week's servings of fruits, vegetables, sweets, salty snacks, and sugar sweetened beverages consumed; micronutrient supplements consumption in the past 30 days; high-dose vitamin A capsule consumption; diarrhea and fever symptoms in the past 7 days; and anthropometric status) (S2 Table).

**Sample size and data analysis**

The target sample size (n = 250) for each target group was based on estimating a 50% prevalence (most conservative estimate) of any MND with a precision of ±7 percentage points, and accounting for 20% loss in sample size due to blood collection failure. Participants without venous blood samples were excluded from analyses involving biomarkers that required serum samples. No imputation method was applied for missing biomarker data because the missingness was not at random but was primarily due to technical challenges in obtaining venous samples. We published our statistical analysis plan (https://osf.io/t3zrn/) before data analysis (R version 4.1.1, R Core Team, Vienna, Austria).

We summarized continuous variables as mean ± SD if normally distributed or median (interquartile range, IQR) if not normally distributed, and categorical variables as frequency and percentages (%). To identify factors associated with anemia, deficiencies of iron (by low SF), zinc, vitamin A, B-12, and folate, and elevated biomarkers of inflammation among WRA and PSC, we first assessed the potential independent variables for associations with each outcome in bivariate models (S3 Table). All potential independent variables that showed a significant association with the outcome (P < 0.05) in the bivariate analyses were included in multivariable generalized linear mixed models (GLMMs). These models also incorporated "district" (Kumbungu: yes/no)" and "setting" (urban community: yes/no) as fixed effects to account for the variations introduced by the cluster selection process, while controlling for the random effects associated with each cluster. We controlled for "time since last meal or drink other than water", "time of day blood was collected", "time of blood centrifugation", and "level of hemolysis in the serum sample" in the zinc deficiency models [58], and child age in all the child models. To address potential multicollinearity, the GLMMs analyzed each outcome variable individually, incorporating only relevant community, household, and individual characteristics, and excluding other outcome variables from the model.

Finally, we compared the observed versus expected prevalence of anemia co-occurring with MND or inflammation by using the Rao-Scott chi-squared test of independence. The expected prevalences, assuming independence between anemia and MND/inflammation, were calculated as the product of the prevalence of anemia and the prevalence of MND or inflammation.

**Results**

The background characteristics of the study participants are summarized in **Table 1**. Of note, 68% of households had participants from 2 or more target groups. Among the households across all groups, the median household size was 11

**Table 1. Background characteristics of participants[1].**

| Background characteristics | LW (n = 243) | WRA (n = 224) | PSC (n = 241) |
|---|---|---|---|
| *Household-level characteristics* | | | |
| Household size | 11 (9, 15)[2] | 11 (8, 15) | 11 (9, 15) |
| Number of children under 5 y in household | 1 (1, 2) | 1 (1, 2) | 1 (1, 2) |
| Highest formal educational level in household | | | |
| None | 59 (25.8)[3] | 50 (22.4) | 67 (28.0) |
| Primary | 36 (15.7) | 30 (13.5) | 32 (13.4) |
| Secondary | 97 (42.4) | 108 (48.4) | 101 (42.3) |
| Higher than secondary | 37 (16.2) | 35 (15.7) | 39 (16.3) |
| Household Food Insecurity Access Scale score[4] | 5 (4, 10) | 5 (3, 10) | 5 (3, 9) |
| Moderate or Severe Food Insecurity[4] | 181 (79.0) | 163 (73.1) | 184 (77.0) |
| Source of drinking water | | | |
| Unimproved[5] | 103 (45.0) | 103 (46.2) | 105 (43.9) |
| Improved[6] | 126 (55.0) | 120 (53.8) | 134 (56.1) |
| Toilet facility | | | |
| Unimproved[7] | 158 (69.0) | 154 (69.1) | 167 (69.9) |
| Improved[8] | 71 (31.0) | 69 (30.9) | 72 (30.1) |
| *Participant-level characteristics* | | | |
| Age (women, y; children, months) | 29.5 ± 6.38[9] | 29.6 ± 9.1 | 41.3 ± 11.4 |
| Women's educational level/Child's level in school | | | |
| None | 165 (72.1) | 154 (69.4) | 151 (63.2) |
| Preschool | 3 (1.3) | 3 (1.4) | 60 (25.1) |
| Primary | 24 (10.5) | 22 (9.9) | 28 (11.7) |
| Secondary | 37 (16.2) | 43 (19.4) | 0 (0.0) |
| Typical week's servings of fruits | 2 (0, 3) | 2 (1, 3) | 2 (0, 2) |
| Typical week's servings of vegetable | 14 (8, 24) | 14 (10, 24) | 5 (2.5, 12) |
| Typical week's servings of sweets [10] | 0 (0, 2) | 0 (0, 2) | 2 (1, 4) |
| Typical week's servings of salty snack[11] | 2 (0, 3) | 2 (0, 3) | 0.5 (0, 2) |
| Typical week's servings of sugar sweetened beverages[12] | 0 (0, 3) | 1 (0, 3) | 0 (0, 3) |
| Consumed micronutrient supplement in past 30 d[13] | 13 (5.7) | 19 (8.8) | 20 (8.5) |
| Consumed micronutrient powder in past 30 d[14] | 2 (0.9) | 0 (0.0) | 3 (1.3) |
| Bouillon intake/d via AME method[15], g | 2.1 (1.5, 3.1) | 1.8 (1.2, 2.5) | 0.9 (0.6, 1.3) |
| Had malaria treatment in the past 4 wk | 11 (4.8) | 18 (8.0) | 20 (8.5) |
| Had ≥ 3 loose stools in 24 h in the past 7 days | 10 (4.4) | 4 (1.8) | 15 (6.3) |
| Had fever in the last 7 days | 65 (28.4) | 64 (28.6) | 78 (32.6) |
| Mid-Upper Arm Circumference, cm | | | 14.8 ± 1.1 |
| Height-for-age z -score (HAZ) | | | −1.3 ± 1.5 |
| Weight-for-height z-score (WHZ) | | | −0.5 ± 0.6 |
| Body Mass Index, kgm$^{-2}$ | 22.5 ± 4.3 | 22.4 ± 3.9 | |
| Underweight (BMI < 18.5 kg/m$^2$) | 12 (5.1) | 21 (9.4) | |

*(Continued)*

**Table 1.** (Continued)

| Background characteristics | LW<br>(n = 243) | WRA<br>(n = 224) | PSC<br>(n = 241) |
|---|---|---|---|
| Stunted (HAZ < −2 SD) | | | 73 (31.5) |
| Wasted (WHZ < −2 SD) | | | 11 (4.7) |

Abbreviations: AME, Adult Male Equivalent; HFIAS, Household Food Insecurity Access Scale; LW, lactating women; MNP, micronutrient powder; PSC, Pre-school children (2–5 y); WRA, women of reproductive age (15–49 y)

1 Participants from different target groups could be recruited from the same household (and therefore some households are represented in multiple columns), but households were not required to have participants from different target groups.

2 All such values are median (interquartile range).

3 All such values are frequency (%).

4 HFIAS score is a continuous measure of the degree of food insecurity based on a set of questions that encompass three domains of food insecurity: (i) anxiety and uncertainty about the household food supply, (ii) insufficient quality, and (iii) insufficient food intake and its physical consequences [23]. Higher values are indicative of higher household food insecurity.

5 Unimproved sources of drinking water include unprotected dug well, unprotected spring, and surface water [24].

6 Improved sources of drinking water include: piped water (into dwelling, compound, yard or plot, public tap/standpipe), tube well/borehole, protected dug well, protected spring, rainwater collection, and packaged or delivered water [24].

7 Unimproved toilet facility disposes of feces in fields, forests, bushes, open water bodies of water, beaches or other open spaces, or with solid waste, a practice known as 'open defecation' [25].

8 Improved toilet facility (hygienically separates human excreta from human contact) include flush or pour flush to piped sewer systems, septic tanks or pit latrines, ventilated improved pit latrines, pit latrines with slabs and composting toilets [25].

9 All such values are Mean ± SD.

10 Sweet servings include sweet snacks such as biscuits, candies or chocolates.

11 Salty servings include salty snacks like crisps and salty crackers.

12 Sugar sweet beverage include sweet beverages sweetened with sugar, such as soft drinks and sugar-sweetened fruit drinks.

13 Any vitamin or mineral supplements not including micronutrient powder mixed with food.

14 Micronutrient powder containing 15 micronutrients per 1 g sachet (https://supply.unicef.org/s0000225.html).

15 The Adult Male Equivalent method was applied to estimate "apparent intake" of bouillon per day [27].

persons, about 26% consisted of members none of whom had any formal education, about 77% reportedly experienced moderate or severe food insecurity, about 45% primarily used an unimproved source of drinking water, and 69% primarily relied on an unimproved toilet facility.

At the individual level, the LW and WRA had nearly the same average characteristics (e.g., average age of about 30 y; about 70% with no formal education; relatively low fruit and vegetable intakes in a typical week, etc.), except for the percentages who had ≥ 3 loose stools (LW, 4.4%; WRA, 1.8%) in 24 h in the past 7 days. The mean body mass index of the WRA was 22 kg m$^{-2}$. Among PSC, the average age was 41 months, most (63%) were not in preschool or primary school, up to 9% (20/241) reportedly consumed a micronutrient supplement or micronutrient powder (3/241) at least once during the past 30 d, and the median number of servings in a typical week was 2 for fruits and 5 for vegetables. Furthermore, about 9% were reportedly treated for malaria in the past 4 wk, while 6% reportedly had diarrhea and 33% reportedly had fever in the past 7 d. The anthropometric assessment showed that more than 31% were stunted, and nearly 5% were wasted.

**Table 2** shows the median (Q1, Q3) concentrations of breast milk retinol (61.2 (45.4, 81.9) nmol/g fat) and vitamin B-12 (240 (191, 347) pmol/L) among the LW. At least 8% had low breastmilk vitamin A (retinol <28 nmol/g fat); the prevalence of low breastmilk B-12 was 77% when using the cut-off value of <362 pmol/L, and 41% when applying the same cut-off (<221 pmol/L) usually used to define low serum or plasma B-12.

**Table 2. Breastmilk vitamin A and vitamin B-12 among lactating women.**

| | Lactating women (n = 231) [1] |
|---|---|
| Vitamin A (retinol) concentration, nmol/g fat | 61.2 (45.4, 81.9) |
| Vitamin B-12 concentration, pmol/L | 240 (191, 347) |
| Low retinol concentration (< 28 nmol/g fat) [41] | 13/165(8) |
| Low B-12 concentration (<362 pmol/L[2] | 179/231 (77) |
| Low B-12 concentration (< 221 pmol/L)[2] | 95/231 (41) |

1 Values are median (interquartile range) or *n*/total *n* (%).

2 We defined low breast milk B-12 concentration using two cut-offs, namely <362 pmol/L [43, 44] and <221 pmol/L often used for serum/plasma vitamin B-12 [45], as there is no universally accepted cut-off for low vitamin B-12 in breast milk [42] and the existing cut-off of <362 pmol/L is not widely used due to potential overestimation of inadequacy, since it was developed using older laboratory methods that may not have measured breast milk vitamin B-12 accurately and was based on a relatively small number of samples [43, 44].

The continuous blood and biochemical outcomes among the WRA and PSC are in **Table 3**. Among WRA, mean ± SD Hb concentration was 123 ± 15 g/L. For iron status, the mean ± SD SF concentration (after adjusting for inflammation) was 21.7 ± 19.3 µg/L, and that for sTfR was 9.6 ± 6.3 mg/L, resulting in BIS of 0.6 ± 4.6 mg/kg. Median UIC was 94 µg/L, indicating "insufficient" iodine intake of the population according to WHO guidelines [40]. Among the PSC, mean ± SD Hb was

**Table 3. Continuous blood and biochemical outcomes among the women of reproductive age and preschool children[1].**

| Variable and concentration | WRA (n = 224) [2] | PSC (n = 241)[2] |
|---|---|---|
| Hemoglobin, g/L | 123 ± 15 [220] | 111 ± 15 [236] |
| *Iron status* | | |
| Serum ferritin, µg/L | 21.7 ± 19.3 [216][3] | 15.6 ± 16.8 [166][3] |
| Soluble transferrin receptor, mg/L | 9.6 ± 6.3 [216] | 13.5 ± 7.8 [166] |
| Body iron stores, mg/kg | 0.6 ± 4.6 [216][3] | −2.3 ± 5.1 [166][3] |
| *Iodine status* | | |
| Urinary iodine concentration, µg/L | 94 (59, 150) [220] | 105 (59, 182) [232] |
| *Vitamin status* | | |
| Serum B-12, pmol/L | 432 ± 224 [209] | 384 ± 196 [141] |
| Erythrocyte folate, nmol/L | 502 ± 216 [208] | |
| Serum folate, nmol/L | 13.8 ± 5.7 [213] | |
| Serum retinol, µmol/L | 1.4 ± 0.4 [202] | 0.9 ± 0.3 [147][4] |
| Retinol binding protein, µmol/L | 1.2 ± 0.4 [216] | 0.7 ± 0.2 [166][4] |
| *Zinc status* | | |
| Serum zinc, µg/dL | 62.7 ± 9.7 [219] | 61.1 ± 8.9 [166][4] |
| *Biomarkers of inflammation* | | |
| alpha-1-acid glycoprotein, g/L | 0.7 ± 0.2 [216] | 0.9 ± 0.3 [166] |
| C-reactive protein, mg/L | 2.5 ± 7.5 [216] | 3.1 ± 7.4 [166] |

Abbreviations: PSC, Pre-school children (2–5 y); WRA, women of reproductive age (15–49 y)

1 Participants from different target groups could be recruited from the same household. We missed venous blood samples from some (29%) of the PSC because of the difficulty of drawing blood from those children.

2 Values are Mean ± SD [n] or Median (Q1, Q3) [n]

3 After adjusting serum ferritin values for inflammation [47].

4 Inflammation adjusted [47].

111 ± 15 g/L; regarding iron status, the mean ± SD SF was 15.6 ± 16.8 µg/L, while that of sTfR was 13.5 ± 7.8 mg/L, resulting in BIS of −2.3 ± 5.1 mg/kg. The median UIC (105 µg/L) indicated "adequate" iodine intake of the population [40].

Table 4 shows the prevalence of anemia, micronutrient deficiency, and elevated biomarkers of inflammation among the WRA and PSC. For WRA, anemia prevalence was 31%, comprising 15% mild, 15% moderate, and 1% severe anemia; iron deficiency prevalence was 45% by SF level, 38% by sTfR, 37% by BIS, and 56% when combining all three indicators; and 19% had IDA based on SF. The prevalence of the other MNDs included < 1% for vitamin A deficiency, 12% for vitamin B-12, 2–12% for folate based on low erythrocyte or serum folate, and 79% for zinc. At least 11% had elevated AGP, 11% had elevated CRP, and 15% had elevated AGP or CRP.

Table 4. Anemia, micronutrient deficiency, and elevated biomarkers of inflammation among the women of reproductive age and preschool children[1].

| Variable | WRA (n = 224)[2] | PSC (n = 241)[2] |
|---|---|---|
| Anemia [48] | | |
| Any anemia (WRA: Hb < 120 g/L; PSC: Hb < 110 g/L) | 68/220 (31) | 85/236 (36) |
| Mild anemia (WRA: Hb 110–119 g/L; PSC: Hb 100–109 g/L) | 33/220 (15) | 45/236 (19) |
| Moderate anemia (WRA: Hb 80–109 g/L; PSC: Hb 70–99 g/L) | 33/220 (15) | 38/236 (16) |
| Severe anemia (WRA: Hb < 80 g/L; PSC: Hb < 70 g/L) | 2/220 (1) | 5/236 (2) |
| ID | | |
| ID by SF (WRA: < 15 µg/L; PSC: < 12 µg/L) [49] | 97/216 (45)[3] | 95/166 (57)[3] |
| ID by sTfR (WRA: > 8.3 mg/L; PSC: > 8.3 mg/L) [36] | 82/216 (38) | 118/166 (71) |
| ID by BIS (WRA: < 0 mg/kg; PSC: < 0 mg/kg) [46] | 80/216 (37)[3] | 101/166 (61)[3] |
| Any ID (WRA: SF < 15 µg/L or sTfR > 8.3 mg/L; PSC: SF < 12 µg/L or sTfR > 8.3 mg/L | 121/216 (56) | 134/166 (81) |
| IDA by SF (WRA: SF < 15 µg/L and Hb < 120 g/L; PSC: SF < 12 µg/L and Hb < 110 g/L) [50] | 40/212 (19)[3] | 41/163 (25)[3] |
| Vitamin A deficiency | | |
| Vitamin A deficiency (SR < 0.70 µmol/L) [52] | 2/202 (1) | 43/147 (29)[4] |
| Vitamin A deficiency (WRA: RBP < 0.52 µmol/L; PSC: RBP < 0.54 µmol/L) [53] | 0/216 (0) | 32/166 (19)[4] |
| Vitamin B-12 deficiency (serum B-12 < 221 pmol/L) [45] | 25/209 (12) | 27/141 (19) |
| Folate deficiency | | |
| "Insufficient" RBC folate (< 748 nmol/L) [54, 55] | 190/208 (91) | |
| Low RBC folate (< 305 nmol/L) [55, 56] | 24/208 (12) | – |
| Low serum folate (serum folate < 7 nmol/L) [55, 56] | 5/213 (2) | – |
| Zinc deficiency [51] | 173/219 (79) | 111/166 (67)[4] |
| Elevated biomarkers of inflammation | | |
| AGP (> 1.0 g/L) [57] | 24/216 (11) | 61/166 (37) |
| CRP (> 5 mg/L) [57] | 24/216 (11) | 28/166 (17) |
| AGP (> 1.0 g/L) or CRP (> 5 mg/L) | 32/216 (15) | 65/166 (39) |

Abbreviations: AGP, alpha-1-acid glycoprotein; BIS, body iron stores; CRP, C-Reactive Protein; Hb, Hemoglobin; ID, iron; PSC, Pre-school children (2–5 y); WRA, women of reproductive age (15–49 y)

deficiency; IDA, iron deficiency anemia; PSC, Pre-school children; RBP, retinol binding protein; SF, serum ferritin; SR, serum retinol; sTfR, soluble transferrin receptor; WRA, women of reproductive age

1 Participants from different target groups could be recruited from the same household. We missed venous blood samples from some (29%) of the PSC because of the difficulty of drawing blood from those children.

2 Values are n/total n (%)

3 After adjusting ferritin values for inflammation [47].

4 Inflammation adjusted [47].

For PSC, anemia prevalence was 36% comprising 19% mild, 16% moderate, and 2% severe anemia; the prevalence of iron deficiency was 57% by SF, 71% by sTfR, 61% by BIS, and 81% when combining all three biomarkers; and 25% had IDA by SF. The prevalence of the other MNDs included 67% for zinc, 19−29% for vitamin A based on low serum retinol or RBP, and 19% for vitamin B-12. At least 37% had elevated AGP, 17% had elevated CRP, and 39% had elevated AGP or CRP.

Results of the bivariate analyses are in S3 Table. The GLMMs results (Table 5) showed that among WRA, the independent variable most frequently associated with anemia and micronutrient deficiency (≥2 associations at P<0.05) was improved source of drinking water (associated with anemia, iron deficiency, and vitamin A deficiency). Specifically, living in households with an improved source of drinking water was associated with a 59% reduction in the odds of having anemia (Adjusted odd ratio, AOR: 0.41; 95% CI: 0.19, 0.87), 52% reduction in the odds of having ID (AOR: 0.48 (0.24, 0.97), and 64% reduction in the odds of having vitamin A deficiency (AOR: 0.36 (0.14, 0.90). Independent variables significantly associated with greater odds of micronutrient deficiency (P<0.05) included living in Kumbungu district for ID [AOR: 2.11 (1.10, 4.04)], food insecurity (higher HFIAS score) for ID [1.09 (1.01, 1.17)], and greater typical week's servings of sugar sweetened beverages (SSBs) for vitamin A deficiency [1.19 (1.06, 1.35)]. None of the independent variables was independently associated with elevated biomarkers of inflammation at P<0.05.

Among PSC, independent variables most frequently associated with anemia and micronutrient deficiency (≥2 associations at P<0.05) were living in rural community, living in the Kumbungu district, height-for-age z-score (HAZ), and reported typical week's servings of fruits. Living in a rural community was associated with lower odds of anemia [AOR: 0.35 (0.19, 0.64) and ID [AOR: 0.42 (0.18, 0.99)], but higher odds of vitamin B-12 deficiency [AOR: 4.07 (1.43, 11.53)]. Living in the Kumbungu district was associated with lower odds of zinc deficiency [0.48 (0.23, 0.99)], but higher odds of vitamin A deficiency [4.40 (1.60, 12.13)]. Higher HAZ was associated with lower odds of anemia [0.74 (0.60, 0.92)] and vitamin B-12 [0.62 (0.4, 0.95)] deficiency. Higher reported number of typical week's servings of fruits was associated with lower odds of anemia: [0.74 (0.60, 0.92)] and vitamin A [0.74 (0.56, 0.99)] deficiency. Other factors significantly associated (P<0.05) with lower odds of anemia or a single micronutrient deficiency included improved sanitation for zinc deficiency [0.35 (0.15, 0.79)], higher age for anemia [0.95 (0.93, 0.98)], and being female for ID [0.38 (0.18, 0.79)]. Factors significantly associated with higher odds of a single micronutrient deficiency included higher reported number of typical week's salty snacks servings for zinc deficiency, and reported fever in the last 7 days for vitamin A deficiency. Only reported fever in the last 7 days [AOR: 2.73 (1.28, 5.82)] and reported malaria treatment in the last 4 wk [AOR: 3.81 (1.03, 14.05)) were associated with elevated biomarkers of inflammation at P<0.05.

Table 6 presents the observed and expected prevalence of co-occurring anemia and MND or inflammation. Among the WRA, the observed prevalence was significantly greater than the prevalence expected by chance alone for anemia co-occurring with ID by SF (19% vs 14%; P=0.002), any ID (23% vs 17%; P=0.001), vitamin A deficiency (1% vs 0.3%; P=0.028), and elevated CRP (5% vs. 3%; P=0.023). Anemia was not associated with the deficiencies of zinc, B-12, folate, elevated AGP or any inflammation.

Among the PSC, the observed prevalence was significantly greater than the prevalence expected by chance alone for anemia co-occurring with ID by SF (25% vs 20%; P=0.006), any iron deficiency (33% vs 28%; P=0.001), elevated AGP (17% vs. 13%; P=0.017), and any inflammation (18% vs. 14%; P=0.042). Anemia was not associated with the deficiencies of zinc, vitamin A, B-12, or elevated CRP.

## Discussion

We found that among LW in this setting, the prevalence of low breastmilk vitamin A was low compared with the prevalence of low breastmilk vitamin B-12. Among non-lactating WRA, vitamin A deficiency was low, but anemia and deficiencies of iron, zinc, vitamin B-12, and folate were high, along with high prevalence of elevated biomarkers of inflammation. Living in a household with an improved source of drinking water was associated with lower odds of anemia, iron deficiency, and

**Table 5. Adjusted odds ratios (95% CI) for factors associated with anemia, micronutrient deficiency, and inflammation[1].**

| Predictors | Anemia[2] | Iron deficiency by ferritin[3] | Zinc deficiency[4] | Vitamin A deficiency[5] | B-12 deficiency[6] | Folate deficiency[7] | Inflammation[8] |
|---|---|---|---|---|---|---|---|
| **WRA** | n = 219 | n = 215 | n = 217 | n = 199 | n = 206 | n = 207 | n = 212 |
| Improved source of drinking water | 0.41* (0.19, 0.87) | 0.48* (0.24, 0.97) | | 0.36* (0.14, 0.90) | | | |
| Living in rural community | 1.08 (0.55, 2.11) | 0.43* (0.23, 0.82) | 1.97 (0.76, 5.13) | 1.44 (0.62, 3.32) | 2.18 (0.76, 6.28) | 0.74 (0.30, 1.79) | 1.27 (0.39, 4.12) |
| Living in the Kumbungu district | 1.45 (0.72, 2.91) | 2.11* (1.10, 4.04) | 0.40 (0.15, 1.09) | 1.72 (0.74, 3.98) | 0.72 (0.25, 2.04) | 0.69 (0.27, 1.76) | 0.80 (0.26, 2.49) |
| HFIAS[9] | | 1.09* (1.01, 1.17) | | | | | 1.08 (0.96, 1.22) |
| Number of children under 5 y in Hh | 0.69* (0.51, 0.94) | | | | 0.66§ (0.41, 1.08) | 0.91 (0.82, 1.01)§ | |
| Hh maximum educ. level = Primary | | | | 0.77 (0.25, 2.4) | | | 4.02§ (0.86, 18.73) |
| Hh maximum educ. level = Secondary | | | | 1.65 (0.7, 3.90) | | | 1.20 (0.28, 5.10) |
| Hh maximum educ. level > secondary | | | | 0.11* (0.01, 0.93) | | | 0.34 (0.04, 2.63) |
| Woman's educ. level = Primary | | | | | | | 2.44 (0.57, 10.5) |
| Woman's educ. level = Secondary | | | | | | | 2.86 (0.59, 13.8) |
| Being married | | | | | 0.52 (0.19, 1.43) | | |
| Apparent bouillon consumption, g/wk[9] | | | 0.74* (0.56, 0.98) | | | | |
| Received vitamin A after most recent birth | | | | | 5.05 (0.65, 39.1) | | |
| Typical week's servings of SSB[9] | | | | 1.19* (1.06, 1.35) | | | |
| Employment status = Home | | | | | | | 0.91 (0.25, 3.32) |
| Employment status = Student | | | | | | | 2.22 (0.51, 9.74) |
| Typical week's servings of vegetables[9] | | | | | | | 1.00 (0.94, 1.06) |
| **PSC 2–5 y[10]** | n = 225 | n = 164 | n = 162 | n = 143 | n = 137 | | n = 165 |
| Living in rural community | 0.35‡ (0.19, 0.64) | 0.42* (0.18, 0.99) | 1.51 (0.7, 3.28) | 0.56 (0.21, 1.49) | 4.07† (1.43, 11.53) | | 0.73 (0.37, 1.47) |
| Living in the Kumbungu district | 1.39 (0.77, 2.5) | 2.17§ (0.91, 5.15) | 0.48* (0.23, 0.99) | 4.40* (1.60, 12.13) | 1.63 (0.64, 4.18) | | 0.79 (0.39, 1.59) |
| Height-for-age z-score[9] | 0.74† (0.60, 0.92) | | | | 0.62* (0.4, 0.95) | | |
| Typical week's servings of fruits [9] | 0.74† (0.60, 0.92) | | | 0.74* (0.56, 0.99) | | | |
| Improved sanitation | | | 0.35* (0.15, 0.79) | | | | |
| Caregiver's educ. level = Preschool | | | | | 0.26§ (0.06, 1.03) | | |
| Caregiver's educ. level = Primary | | | | | 0.38 (0.04, 3.59) | | |

*(Continued)*

**Table 5.** (Continued)

| Predictors | Anemia[2] | Iron deficiency by ferritin[3] | Zinc deficiency[4] | Vitamin A deficiency[5] | B-12 deficiency[6] | Folate deficiency[7] | Inflammation[8] |
|---|---|---|---|---|---|---|---|
| Age, y[9] | 0.95[‡] (0.93, 0.98) | 0.98 (0.95, 1.01) | 0.97[§] (0.93, 1.00) | 1.01 (0.97, 1.05) | 0.99 (0.93, 1.04) | | 0.97[§] (0.94, 1.00) |
| Child sex=Female | | 0.38[*] (0.18, 0.79) | | | | | |
| Typical week's servings of salty snack[9] | | | 1.39[*] (1.05, 1.83) | | | | |
| Had fever in the last 7 days | | | | 3.17[*] (1.27, 7.91) | | | 2.73[*] (1.28, 5.82) |
| Had malaria treatment in last 4 wk | | | | | | | 3.81[*] (1.03, 14.05) |

Abbreviations: AGP, alpha-1-acid glycoprotein; CRP, C-reactive protein; HFIAS, Household Food Insecurity and Access Score; Hh, Household head; IDA, iron deficiency anemia; SSB, sugar-sweetened beverage; WRA, women of reproductive age.

1 Participants from different target groups could be recruited from the same household. Estimates are odds ratios and represent the relative change in the expected odds of the outcome associated with a one-unit change in the independent variable, by generalized linear mixed-model. Only predictors significantly associated with each outcome at 0.05 level of significance in bivariate analyses were included in the multivariable models presented here. § $p < 0.1$, * $p < 0.05$, † $p < 0.01$, ‡ $p < 0.001$.

2 Anemia was defined as Hemoglobin (Hb) < 120 g/L in WRA (n = 219) and Hb < 110 g/L in children 2–5 y of age (n = 225).

3 ID by ferritin was defined as the presence of ferritin (SF) < 15 µg/L in WRA (n = 215) and SF < 12 µg/L in children 2–5 y of age (n = 164). SF concentration was adjusted for inflammation as described by the World Health Organization [49].

4 Zinc deficiency was defined as serum zinc < 70 µg/dL when fasting or < 66 µg/dL when non-fasting in WRA (n = 217) and serum zinc < 65 µg/dL in children 2–5 y of age (n = 162). Serum zinc was adjusted for inflammation.

5 Vitamin A deficiency was defined as serum retinol < 1.05 µmol/L in WRA (n = 199) and < 0.70 µmol/L in children 2–5 y of age (n = 143). Serum retinol was adjusted for inflammation in children.

6 Vitamin B-12 deficiency was defined as serum vitamin B-12 concentration < 221 pmol/L in WRA (n = 206) and children 2–5 y of age (n = 137).

7 Folate deficiency was defined as RBC folate < 305 nmol/L in WRA (n = 207).

8 Inflammation was defined as CRP > 5 mg/L or AGP > 1.0 g/L [57] in WRA (n = 212) and children 2–5 y of age (n = 165).

9 Variables were considered as continuous variables.

10 We missed venous blood samples from some (29%) of the PSC because of the difficulty of drawing the blood from those children.

vitamin A deficiency. Living in the Kumbungu district, food insecurity, and higher consumption of SSBs were significantly associated with higher odds of MNDs, including ID and vitamin A deficiency. Anemia was associated with deficiencies of iron and vitamin A, as well as elevated CRP levels, but was not associated with deficiencies of zinc, B-12, and folate, nor with elevated AGP levels.

Among the PSC, the prevalence of anemia and deficiencies of iron, zinc, vitamin A, and vitamin B-12 were high, along with a high prevalence of elevated biomarkers of inflammation. Living in a rural area or the Kumbungu district was linked to mixed outcomes; rural residence was linked to lower odds of anemia and iron deficiency but higher odds of vitamin B-12 deficiency, while Kumbungu residence was linked to lower odds of zinc deficiency but higher odds of vitamin A deficiency. Higher HAZ and higher fruit consumption were consistently associated with lower odds of anemia and vitamin A or vitamin B-12 deficiency. Anemia co-occurred with iron deficiency and elevated AGP, but not with the other outcomes.

This study had several strengths: we determined outcomes of public health importance in a part of Ghana often considered as a "hotspot" of poor maternal and child nutrition. The results presented are not typically available in the Ghana Demographic and Health Survey or the Multiple Indicator Cluster Survey reports, and they offer a new understanding of factors associated with anemia and MNDs among the target groups in the two districts. We used robust statistical approaches, including GLMMs controlling for the random effect of cluster. Giving priority to recruiting multiple target group members from the same household offered several advantages, including providing a broader picture of micronutrient

**Table 6. Comparison of observed prevalence of co-occurring anemia and micronutrient deficiency or inflammation with the prevalence expected by chance[1].**

| Co-occurring anemia[3] and micro-nutrient deficiency or inflammation | Women of reproductive age (n = 224) | | | Children 2–5 years of age (n = 241) | | |
|---|---|---|---|---|---|---|
| | Observed co-occurrence, % | Expected co-occurrence, % | P[2] | Observed co-occurrence, % | Expected co-occurrence, % | P[2] |
| Anemia + iron deficiency by SF[4] | 18.9 | 13.8 | 0.002 | 25.2 | 20.2 | 0.006 |
| Anemia + any iron deficiency[5] | 22.6 | 17.2 | 0.001 | 33.1 | 28.3 | 0.001 |
| Anemia + zinc deficiency[6] | 27.4 | 24.6 | 0.06 | 20.1 | 23.3 | 0.06 |
| Anemia + vitamin A deficiency[7] | 1.0 | 0.3 | 0.028 | 11.0 | 9.5 | 0.40 |
| Anemia + vitamin B-12 deficiency[8] | 3.9 | 4.0 | 0.96 | 7.2 | 6.7 | 0.71 |
| Anemia + folate deficiency[9] | 4.8 | 3.7 | 0.20 | – | – | – |
| Anemia + elevated AGP[10] | 3.3 | 3.1 | 0.86 | 17.2 | 12.9 | 0.017 |
| Anemia + elevated CRP[11] | 5.2 | 3.1 | 0.023 | 4.9 | 6.0 | 0.43 |
| Anemia + any inflammation (elevated AGP or CRP)[12] | 5.7 | 4.4 | 0.34 | 17.8 | 13.7 | 0.042 |

Abbreviations: AGP, alpha-1-acid glycoprotein; CRP, high C-Reactive Protein; SF, serum ferritin.

1 Participants from different target groups could be recruited from the same household.

2 P-values compare the observed to the expected prevalences assuming that the null hypothesis (no association between the variables) was true, by Rao Scott chi-squared tests accounting for study design.

3 Anemia was defined as Hb < 120 g/L in women and Hb < 110 g/L in children.

4 Iron deficiency by ferritin was defined as SF < 15 µg/L in women and SF < 12 µg/L in children, after adjusting SF concentration for inflammation [49].

5 Any iron deficiency defined as SF < 15 µg/L and/or sTfR > 8.3 mg/L in women, and SF < 12 µg/L and/or serum soluble transferrin (sTfR) > 8.3 mg/L in children after adjusting SF concentration for inflammation [49].

6 Zinc deficiency was defined as serum zinc < 70 µg/dL when fasting or < 66 µg/dL when non-fasting in women, and serum zinc < 65 µg/dL in children. Zinc concentration was adjusted for inflammation (AGP and CRP) in children [47].

7 Vitamin A deficiency was defined as serum retinol < 0.70 µmol/L in WRA and children [52]. Retinol concentration was adjusted for inflammation (AGP and CRP) in children [47].

8 Vitamin B-12 was defined as serum vitamin B12 concentration < 221 pmol/L) in women and children.

9 Folate deficiency in women was defined as RBC folate < 305 nmol/L.

10 AGP concentration >1.0 g/L.

11 CRP concentration >5 mg/L.

12 AGP > 1.0 g/L or CRP > 5 mg/L.

status across multiple physiological groups, reducing the variability in outcomes caused by shared household factors (e.g., socio-economic status and dietary intake) thereby saving costs, and improving the efficiency of data collection. A weakness of the study was the selection of clusters from only the parts of the Tolon district easily accessible at the time; we also missed venous blood samples from some of the PSC (29%) because of the difficulty of drawing the blood. It is unlikely, however, that these would limit the generalizability of our results across the districts, which were generally homogenous in terms of geographical characteristics, economic status, and dietary practices [59,60], and the flooding that made some areas in the Tolon district difficult to access was short-lived. Given that the associations between living in a rural community and micronutrient deficiency were only significant for some outcomes (i.e., iron deficiency in WRA and PSD, anemia in PSC only, and vitamin B-12 deficiency in PSC only), the exclusion of some rural clusters in the Tolon district may have introduced limited bias; however, any such bias is likely to be small given the overall geographic and socioeconomic homogeneity of the district.

Our reliance on self-report for dietary data collection may have introduced some bias; however, the use of show-cards with photos of local food items and typical serving sizes likely improved reporting accuracy and consistency. While the FFQ focused on describing consumption patterns of selected food groups, the primary limitation is that we cannot

determine whether the associations observed with MNDs reflect underlying biological mechanisms or whether the dietary patterns serve as markers for other unmeasured factors.

We do not know of any recent or current anemia and MND prevalence data for the Tolon and Kumbungu districts with which to compare our results for the WRA and PSC, except for the anemia prevalence for the Northern Region (women: 48.4%; PSC: 69.4%) from the 2022 Ghana DHS [11] and those for the "Northern Belt" (comprising the Northern, Upper East, and Upper West regions) from the 2017 GMS [12]. In the "Northern Belt", the GMS reported 27.6% anemia, 21.5% iron deficiency (by low SF), and 15.4% IDA, as well as 4.1% vitamin A (RBP < 0.70 µmol/L), 13.5% vitamin B-12 (serum B12 < 150 pmol/L), and 50.8% folate (by serum folate <10 nmol/L) deficiency rates among WRA, and 53.2% anemia, 39.6% iron deficiency (by low SF), 29.0% IDA, and 30.6% vitamin A (by RBP < 0.70 µmol/L) deficiency rates in PSC. Our results generally tracked well with those from the 2017 GMS, except for iron and folate deficiencies in the WRA and anemia and iron deficiency in PSC. Such differences may be expected, given differences in the survey samples and the sampling methodologies. While we did not statistically compare our estimates with those from the 2017 GMS due to obvious differences such as survey design and sampling frames, such descriptive comparisons still offer useful context for interpreting the prevalence patterns observed in our study. Possible explanations to the relatively low prevalence of vitamin A deficiency among WRA compared with relatively high prevalence in PSC might include more diverse diets (including vitamin A-fortified cooking oil consumption) and lower prevalence of infection (as indicated by lower prevalence of elevated AGP and/or CRP, Table 4) among WRA compared with PSC.

The high prevalence of anemia and MNDs among WRA and PSC in this study is consistent with data from other low- and middle-income countries [2, 3]. For instance, the anemia prevalence rates observed in our study (31% in WRA; 36% in PSC) align with the global estimates of 30% among WRA and 40% among PSC [2]. These rates are relatively high compared with those reported for high-income countries, such as the United States, where anemia prevalence was estimated at 11.8% among WRA and 6.1% among PSC in 2019 [61], while the prevalence of MNDs, including folate (0%), iron (22%), zinc (14%) among WRA in 2022 [5] were considerably lower.

In Ghana, the high prevalence of anemia and MNDs may be attributed to various factors, including inadequate dietary intake [62], poor bioavailability of dietary iron [63], poor access to fortified foods or supplements [12], frequent infections [64], and malaria [65], along with increased requirements during childbearing years [66] or childhood [67]. We did not analyze water and food samples for micronutrients and heavy metals, as some of these metals, including calcium [68] and cadmium [69], could potentially influence iron absorption and anemia risk. Future studies may benefit from including these assessments.

The observed significant association between improved source of drinking water and lower odds of anemia, iron deficiency, and vitamin A deficiency among WRA was expected. Access to improved drinking water is often linked to better health outcomes, as it reduces the incidence of waterborne diseases and infections that can cause blood loss and/or impair nutrient absorption, leading to anemia and MND [70, 71]. For instance, in an analysis of Demographic and Health Survey data of 10 eastern African countries [72], unimproved source of drinking water was associated with a higher prevalence ratio of anemia.

The significant associations between living in the Kumbungu district, higher food insecurity, and higher consumption of SSBs with higher odds iron and vitamin A deficiencies in WRA may be attributed to several factors, including differences in food access between districts, negative impact of food insecurity on dietary quality, and nutrient displacement from SSB. Food insecurity is known to limit dietary diversity and reduces intake of micronutrient-rich foods, leading to inadequate nutrient intake [73,74]. Households experiencing food insecurity often consume fewer animal-source foods and fresh produce, which are important for iron and folate supply [75,76]. High consumption of SSBs may be linked to lower intake of essential nutrients, as these beverages often provide calories with minimal nutritional value, reducing the consumption of more nutritious options [77]. It is possible there are other explanations for the observed associations [78,79]. Our findings call for food security interventions that improve both the availability and quality of food to prevent MNDs.

 

Among PSC, several possible explanations may account for why rural residence was associated with lower odds of anemia and ID but higher odds of vitamin B-12 deficiency, while living in the Kumbungu district was linked to lower odds of zinc deficiency but higher odds of vitamin A deficiency. It is conceivable that rural residents may have greater access to fresh produce and traditional diets rich in iron, which may explain the lower odds of anemia and ID, while limited access to animal products, which are primary sources of vitamin B-12, might contribute to the higher odds of vitamin B-12 deficiency. However, our previous analysis [80] showed that the proportion of PSC who consumed fruits in a typical week was significantly greater in the urban areas (82%) than in the rural areas (68%), while vegetable consumption was not associated with urban or rural residence. It is unclear how differences in dietary patterns between the two districts, including the consumption of foods rich in zinc, such as groundnut soup (4.2 mg Zn/100 g) and *tuo-zaafi* (4.0 mg Zn/100 g) [81], may have contributed to the lower odds of zinc deficiency, but higher odds of vitamin A deficiency in the Kumbungu district. The findings that higher HAZ and higher typical week's servings of fruits consumed were associated with lower odds of anemia and vitamin A or vitamin B-12 deficiency may reflect better overall nutrition and health status. Higher fruit consumption might reflect a more diverse diet in general, contributing to reduced odds of anemia and MND.

The co-occurrence of anemia with MNDs, such as iron and vitamin A deficiency, observed among WRA and PSC is consistent with the evidence that MNDs often co-exist [14,82]. ID is a primary cause of anemia, but deficiencies in vitamin A, zinc, and folate can also contribute to anemia by impairing immune function, reducing nutrient absorption, and affecting red blood cell production [83,84]. Additionally, the co-occurrence of anemia with elevated AGP or CRP, supports the "anemia of inflammation" hypothesis where inflammation disrupts iron metabolism and contributes to anemia [85]. These findings support the need for interventions that address multiple MNDs and inflammation to reduce anemia prevalence [86] in this setting.

## Conclusions

We observed a high prevalence of various micronutrient deficiencies among WRA and PSC in the two districts despite the existence of micronutrient intervention programs in Ghana, including the mandatory fortification of edible oil with vitamin A and wheat flour with vitamin A, iron, zinc, folic acid, and vitamin B12 [87].

Our findings suggest that current national fortification programs targeting edible oil and wheat flour may not sufficiently reach or impact all subpopulations, particularly in rural or underserved areas. While the use of multiple micronutrient powder in home-fortification [88] might contribute to improving micronutrient intakes of PSC in this setting, its long-term sustainability remains uncertain [89]. Thus, national stakeholders should consider reviewing and potentially expanding the scope of existing fortification efforts. In particular, bouillon fortification has been proposed as a possible strategy to increase dietary micronutrient adequacy among individuals at risk of micronutrient deficiency [90,91]. Bouillon cubes are widely consumed in Ghana [92] and may serve as an effective additional vehicle for delivering key micronutrients and improving coverage. In our previous analysis [93], 99% of 369 surveyed households reported ever using bouillon in cooking, and 77% reported typically using it two or more times per day. Our data can guide which micronutrients are needed, and help identify the population groups that may benefit most. Based on the results of this study, an efficacy trial of the impact of multiple micronutrient-fortified bouillon on micronutrient status was implemented in these same study districts [16].

To reduce micronutrient deficiencies in this and similar settings, potential strategies may include introducing additional delivery vehicles, such as multiple micronutrient-fortified bouillon cubes. Given the observed associations between improved drinking water and reduced risk of anemia and micronutrient deficiencies, ensuring access to safe drinking water may provide additional benefits. Policymakers and program implementers should consider the potential benefits and costs of these multi-sectoral approaches when designing interventions.

## Supporting information

**S1 Table. Definitions and cut-offs used for the binary outcomes.**
(DOCX)

**S2 Table. Potential independent variables for bivariate analysis of anemia, micronutrient deficiencies, and elevated biomarkers of inflammation among women of reproductive age and preschool children.**
(DOCX)

**S3 Table. Bivariate analysis of potential independent variables for anemia, micronutrient deficiency, and inflammation among women of reproductive age and preschool children.**
(DOCX)

## Acknowledgments

We thank the study participants and their families for their cooperation; the Project staff, including Mr. Yakubu Adams, the office personnel, and field workers for their dedication; Mr. Musah Abdul-Karim and the staff at the district office of the Ministry of Agriculture in Tolon for sharing space during data collection; Dr. Victor Adongo at the Tamale Teaching Hospital for assistance with sample collection and storage; the UCSF MLK Cores Research Facility for support with zinc sample analysis; Drs. Lindsay Allen, Setareh Shahab-Ferdows, and Daniela Hampel (University of California, Davis) for support with the vitamin B12 analysis; Shameem Jabbar and Drs. Christine Pfeiffer and Mindy Zhang (CDC) for support with the folate analysis; and Dr. Rochelle Werner for assistance with the milk and serum retinol analysis.

The authors' responsibilities were as follows: SA-A, KRW, MJH, SMK, CDA, KPA, SAV, and RES conceptualized the research; SA-A, KRW, MJH, SMK, CDA, JND, ERB, AF, XT, KPA, SAV, and RES developed the methodology; SA-A, KRW, MJH, SMK, KPA, SAV, and RES were responsible for project administration; SA-A, KRW, MJH, SMK, JND, ERB, AF, KPA, SAV, and RES provided supervision; SA-A, KRW, CDA, and RES drafted the manuscript; and SA-A, KRW, MJH, SMK, CDA, JND, ERB, AF, XT, KPA, SAV, and RES reviewed and edited the manuscript. All authors read and approved the final version of the manuscript and accepted final responsibility for its content. Authors declare that they have no conflict of interest. The funder of the study had no role in the study design, data collection, analysis, interpretation, or preparation of the manuscript.

## Author contributions

**Conceptualization:** Seth Adu-Afarwuah, Sika M. Kumordzie, K. Ryan Wessells, Charles D. Arnold, Stephen A. Vosti, Marjorie J. Haskell, Katherine P. Adams, Reina Engle-Stone.

**Data curation:** Charles D. Arnold, Xiuping Tan.

**Formal analysis:** Charles D. Arnold, Xiuping Tan.

**Methodology:** Seth Adu-Afarwuah, Sika M. Kumordzie, K. Ryan Wessells, Charles D. Arnold, Xiuping Tan, Ahmed Fuseini, Stephen A. Vosti, Marjorie J. Haskell, Emily R. Becher, Jennie N. Davis, Katherine P. Adams, Reina Engle-Stone.

**Project administration:** Seth Adu-Afarwuah, Sika M. Kumordzie, K. Ryan Wessells, Charles D. Arnold, Stephen A. Vosti, Marjorie J. Haskell, Reina Engle-Stone.

**Supervision:** Seth Adu-Afarwuah, Sika M. Kumordzie, K. Ryan Wessells, Ahmed Fuseini, Stephen A. Vosti, Marjorie J. Haskell, Emily R. Becher, Jennie N. Davis, Katherine P. Adams, Reina Engle-Stone.

**Writing – original draft:** Seth Adu-Afarwuah, K. Ryan Wessells, Charles D. Arnold, Marjorie J. Haskell, Jennie N. Davis, Reina Engle-Stone.

**Writing – review & editing:** Seth Adu-Afarwuah, Sika M. Kumordzie, K. Ryan Wessells, Charles D. Arnold, Xiuping Tan, Ahmed Fuseini, Stephen A. Vosti, Marjorie J. Haskell, Emily R. Becher, Jennie N. Davis, Katherine P. Adams, Reina Engle-Stone.

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
