## [Decision Letter · Decision Letter 0]

Mar 29 2025

PONE-D-24-60529Anemia, micronutrient deficiency, and elevated biomarkers of inflammation among women and children in two districts in the Northern Region of Ghana: A pilot studyPLOS ONE

Dear Dr. Adu-Afarwuah,

Thank you for submitting your manuscript to PLOS ONE. After careful consideration, we feel that it has merit but does not fully meet PLOS ONE’s publication criteria as it currently stands. Therefore, we invite you to submit a revised version of the manuscript that addresses the points raised during the review process.

Although manuscript is well-written, the authors must clarify the data availability, as this concern was also raised by the reviewers.

We look forward to receiving your revised manuscript.

Kind regards,

Hansani Madushika Abeywickrama, Ph.D.

Academic Editor

PLOS ONE

Journal Requirements:

“This work was supported, in whole or in part, by a grant from Helen Keller International (66504-UCD-01), through support from the Bill & Melinda Gates Foundation (INV-007916), to the University of California, Davis. Under the grant conditions of the Foundation, a Creative Commons Attribution 4.0 Generic License has already been assigned to the Author Accepted Manuscript version that might arise from this submission.”

5. Please include captions for your Supporting Information files at the end of your manuscript, and update any in-text citations to match accordingly. Please see our Supporting Information guidelines for more information: http://journals.plos.org/plosone/s/supporting-information .

**Additional Editor Comments:**

The authors state that the study areas were "randomly selected," yet they also mention selecting clusters in the Tolon district based on accessibility. This raises questions about the randomness of the selection process. The authors should provide a clear justification for this methodological choice to ensure transparency and consistency in their study design.

Reviewers' comments:

Reviewer's Responses to Questions

**Comments to the Author**

1. Is the manuscript technically sound, and do the data support the conclusions?

Reviewer #1: Yes

Reviewer #2: Yes

Reviewer #3: Partly

Reviewer #4: Yes

2. Has the statistical analysis been performed appropriately and rigorously? 

Reviewer #1: N/A

Reviewer #2: Yes

Reviewer #3: Yes

Reviewer #4: I Don't Know

3. Have the authors made all data underlying the findings in their manuscript fully available?

Reviewer #1: Yes

Reviewer #2: Yes

Reviewer #3: No

Reviewer #4: Yes

4. Is the manuscript presented in an intelligible fashion and written in standard English?

Reviewer #1: Yes

Reviewer #2: Yes

Reviewer #3: Yes

Reviewer #4: Yes

5. Review Comments to the Author

Reviewer #1: Thank you for the opportunity to review this manuscript.

This manuscript is a valuable contribution to understanding anemia and micronutrient deficiencies in Northern Ghana. The manuscript exhibits undeniable strengths, including the robust application of generalized linear mixed models to account for clustering and comprehensive reporting of results. The authors effectively connect their findings to broader public health implications, such as fortification interventions.

Specific areas for improvement:

I. Major:

o Address how missing venous blood samples for 29% of preschool children were handled. Were these participants excluded from analyses, or were imputation techniques applied?

o It is unclear if all raw data (e.g., anonymized participant-level data) are included or easily accessible. The phrase “data will be made available three years after data collection” does not comply with the requirement for immediate availability upon publication. Ensure immediate public access to all data, or explicitly justify any restrictions and provide alternative means for researchers to access restricted data.

II. Minor:

o While the study compares findings to the 2017 Ghana Micronutrient Survey, it lacks statistical comparisons to determine the significance of differences.

o Address potential biases introduced by this approach, particularly in densely populated or inaccessible areas.

o Clusters in the Tolon district were chosen based on accessibility. Discuss how this may limit generalizability and how these biases were mitigated.

o Expand on implications for policy and practice. For example, how can findings inform national micronutrient fortification programs?

o Compare results to international data for broader relevance.

o Emphasize actionable recommendations in the conclusion, such as advocating for specific interventions like water sanitation or fortified foods.

Reviewer #2: DearAuthor;

It is a well-planned study and the number of regions-cases evaluated is sufficient. The aims of the study are emphasized in detail. The results are reported in accordance with the aims. The findings are as expected. However, the results of the study will also be a guide for low-income countries. Anemia is a global problem in childhood and lactation group. It has emphasized the problems of micronutrient deficiency and anemia in this group with its results and has constructive suggestions in line with the results. Current results may guide larger studies.

Best regards

Reviewer #3: Paper Accepted

And anyone who has conducted human research knows what enormous effort was made to gain permission to participate, as well as collect anthropometric measurements and biological samples.

Minor Revisions

Include limitations of Food Frequency Questionnaire (FFQ) to predict actual nutrient intake. A number of studies have shown r-values at or below 0.3 for predicting actual nutrient intake.

Over the years, serving sizes of particular foods (for example fruits and vegetables) have varied. A table with serving sizes shown could be added to the paper's supplemental information. Additionally, in this study, this tool limited its questions to include fruits, vegetables, salty snacks, sweetened snacks, and sugar-sweetened beverages and did not appear to include protein sources. Based on the high level of zinc inadequacy, providing a supplemental list of protein food sources to the questionnaire would seem reasonable.

Because one of the goals of the study is to determine the potential to use bouillon cubes for micronutrient distribution, it would have been helpful to see what percent of the surveyed households presently use bouillon cubes or were willing to incorporate this into their diet regularly.

Location of raw data should be indicated in manuscript.

1) Is the manuscript technically sound, and do the data support the conclusions?

A research study is only as good as its methods. The methods described in this study provide a thorough guide to what was done making it possible to repeat the study. This is a rigorous, with appropriate controls, replication, and sample sizes. Conclusions drawn on anemia and MND are appropriately based on the data presented.

This study's methods are generally accepted as valid. The food frequency questionnaire (FFQ) is often used to estimate the nutritional adequacy of the diet. Sadly, this often used tool has a relatively low correlation coefficient when compared to a 24-hour recall and can lead to misinterpretations. A number of studies have shown r-values at or below 0.3 for predicting actual nutrient intake.

2) Has the statistical analysis been performed appropriately and rigorously?

Based on data presented, it would appear that statistical data is both appropriate and rigorous.

3) Have the authors made all data underlying the findings in their manuscript fully available?

The supplemental data appears to contain more analyses rather than raw data. Location of raw data should be indicated in manuscript.

4) Is the manuscript presented in an intelligible fashion and written in standard English?

Manuscript is well written.

5) As a pilot study, the paper seems to be the first paper of a 2-part study. Follow-up study listed is reference 27.

Benefits to Science:

See comments above regarding methods.

It was refreshing to see the full complement of micronutrients chosen to evaluate levels of anemia and iron deficiency, as well as inflammation indices. The use of well-established labs to analyze the samples also provides a level of confidence in appropriate laboratory technique.

Discussion

The discussion is the weakest part of the paper. The confusing MND information regarding relationship between FFQ and MND may well be due to the low correlation coefficient when compared to a 24-hour recall, this limitation should be included.

Too often sugary and salty foods are thought to be the cause of malnutrition. My 20 years of clinical observations are that these food choices are instead the result of the nutrient inadequacies rather than the cause.

Also, it might have been helpful to have water samples analyzed for micronutrients and heavy metals that could interfere with iron absorption.

Bottom-line

Even with the paper’s limitations, the well detailed methodology is of great benefit for those planning to do human research.

Reviewer #4: Researchers have found that anemia and micronutrient deficiencies are common in the Tolon and Kumbungu districts. This study looked again at these issues in lactating women (LW), women of reproductive age (WRA), and preschool children (PSC) to help test fortified bouillon cubes. They wanted to identify what factors contribute to anemia, micronutrient deficiencies, and inflammation among WRA and PSC. They also examined how often anemia occurs along with these other issues. Even though Ghana has programs to improve micronutrient intake, many WRA and PSC in these districts still face high rates of deficiencies. The researchers suggest that more strategies are needed to reduce these deficiencies in these areas and similar ones.

This is a study with valuable results. All sections of the article are well-written. Below are several comments for your consideration:

Comment 1:

In the introduction, please define micronutrients and explain which factors are considered to be micronutrients. Additionally, clarify why you selected certain micronutrients over others for this study.

Comment 2:

At the end of the introduction, is the following sentence accurate?

(Whether the co-occurrence of anemia with MNDs or inflammation significantly contributed to anemia prevalence.)

Comment 3:

Discussion section, line 483: The high prevalence of anemia and MNDs among WRA and PSC in this study is consistent with data from other low- and middle-income countries.

Besides addressing low- and middle-income countries, compare your results with those of similar studies conducted in high-income countries.

6. PLOS authors have the option to publish the peer review history of their article (what does this mean? ). If published, this will include your full peer review and any attached files.

**Do you want your identity to be public for this peer review?** For information about this choice, including consent withdrawal, please see our Privacy Policy .

Reviewer #1: No

Reviewer #2: No

Reviewer #3: No

Reviewer #4: No

---

## [Author Response · Author response to Decision Letter 1]

22 Apr 2025

Please see the attached 'Response to Reviewers' document

---

## [Decision Letter · Decision Letter 1]

Anemia, micronutrient deficiency, and elevated biomarkers of inflammation among women and children in two districts in the Northern Region of Ghana: A pilot study

PONE-D-24-60529R1

Dear Dr. Adu-Afarwuah,

We’re pleased to inform you that your manuscript has been judged scientifically suitable for publication and will be formally accepted for publication once it meets all outstanding technical requirements.

Kind regards,

Hansani Madushika Abeywickrama, Ph.D.

Academic Editor

PLOS ONE

Additional Editor Comments (optional):

Reviewers' comments:

Reviewer's Responses to Questions

**Comments to the Author**

1. If the authors have adequately addressed your comments raised in a previous round of review and you feel that this manuscript is now acceptable for publication, you may indicate that here to bypass the “Comments to the Author” section, enter your conflict of interest statement in the “Confidential to Editor” section, and submit your "Accept" recommendation.

Reviewer #3: All comments have been addressed

2. Is the manuscript technically sound, and do the data support the conclusions?

Reviewer #3: Yes

3. Has the statistical analysis been performed appropriately and rigorously? 

Reviewer #3: Yes

4. Have the authors made all data underlying the findings in their manuscript fully available?

Reviewer #3: Yes

5. Is the manuscript presented in an intelligible fashion and written in standard English?

Reviewer #3: Yes

6. Review Comments to the Author

Reviewer #3: The revisions made were thoughtful and thorough and added greatly to the manuscript. This study is a great example of researcher and institution collaborations resulting in effective research.

7. PLOS authors have the option to publish the peer review history of their article (what does this mean? ). If published, this will include your full peer review and any attached files.

**Do you want your identity to be public for this peer review?** For information about this choice, including consent withdrawal, please see our Privacy Policy .

Reviewer #3: No

---

## [Editor Report · Acceptance letter]

PONE-D-24-60529R1

PLOS ONE

Dear Dr. Adu-Afarwuah,

I'm pleased to inform you that your manuscript has been deemed suitable for publication in PLOS ONE. Congratulations! Your manuscript is now being handed over to our production team.

Kind regards,

on behalf of

Dr. Hansani Madushika Abeywickrama

Academic Editor

PLOS ONE